# LEX-GAN: Layered Explainable Rumor Detector Based on Generative Adversarial Networks

## Abstract

Social media have emerged to be increasingly popular and have been used as tools for gathering and propagating information. However, the vigorous growth of social media contributes to the fast-spreading and far-reaching rumors. Rumor detection has become a necessary defense. Traditional rumor detection methods based on hand-crafted feature selection are replaced by automatic approaches that are based on Artificial Intelligence (AI). AI decision making systems need to have the necessary means, such as explainability to assure users their trustworthiness. Inspired by the thriving development of Generative Adversarial Networks (GANs) on text applications, we propose LEX-GAN, a GAN-based layered explainable rumor detector to improve the detection quality and provide explainability. Unlike fake news detection that needs a previously collected verified news database, LEX-GAN realizes explainable rumor detection based on only tweet-level text. LEX-GAN is trained with generated non-rumor-looking rumors. The generators produce rumors by intelligently inserting controversial information in non-rumors, and force the discriminators to detect detailed glitches and deduce exactly which parts in the sentence are problematic. The layered structures in both generative and discriminative model contributes to the high performance. We show LEX-GAN's mutation detection ability in textural sequences by performing a gene classification and mutation detection task.

## 1 Introduction

Sequential synthetic data generation such as generating text and images that are indistinguishable to human generated data have become an important problem in the era of Artificial Intelligence (AI). Generative models, e.g., Variational AutoEncoders (VAEs) (Kingma & Welling, 2013), Generative Adversarial Networks (GANs) (Goodfellow et al., 2014), Recurrent Neural Networks (RNNs) with Long Short-Term Memory (LSTM) cells (Hochreiter & Schmidhuber, 1997), have shown outstanding generation power of fake faces, fake videos, etc. Consequently, we require discriminative models capable of detecting AI-generated fake data with explainability in order to manage the malicious viral information (Knight, 2019). A black box decision maker without explainability that does not shed light into how the decision are made may hence lose the trust of its users.

GANs estimate generative models via an adversarial training process (Goodfellow et al., 2014). Powerful real-valued generators have found applications in image and video generation. However, GANs face challenges when the goal is to generate sequences of discrete tokens such as text (Yu et al., 2017). Given the discrete nature of text, backpropagating the gradient from the discriminator to the generator becomes infeasible (Fedus et al., 2018). Training instability is a common problem of GANs, especially those with discrete settings. Unlike image generation, the autoregressive property in text generation exacerbates the training instability since the loss from discriminator is only observed after a sentence has been generated completely (Fedus et al., 2018).

In addition to the recent development in GAN-based text generation, discriminator-oriented GAN-style approaches are proposed for detection and classification applications, such as rumor detection (Ma et al., 2019). Differently from the original generator-oriented GANs, discriminator-oriented GAN-based models take real data as input to the generator instead of noise. Hence fundamentally,

the detector may get high performance through the adversarial training technique. Current adversarial training strategies improve the robustness against adversarial samples. However, these methods lead to reduction of accuracy when the input samples are clean (Raghunathan et al., 2019).

On a related note, social media and micro-blogging have become increasingly popular (Yazdanifard et al., 2011; Viviani & Pasi, 2017). The convenient and fast-spreading nature of micro-blogs fosters the emergence of various rumors. Commercial giants, government authorities, and academic researchers take great effort in diminishing the negative impacts of rumors (Cao et al., 2018). Rumor detection has been formulated into a binary classification problem by a lot of researchers. Traditional approaches based on hand-crafted features describe the distribution of rumors (Castillo et al., 2011; Kwon et al., 2013). More recently, Deep Neural Network (DNN)-based methods extract and learn features automatically, and achieve significantly high accuracy on rumor detection (Chen et al., 2018). Generative models have also been used to improve the performance of rumor detectors (Ma et al., 2019). However, binary rumor classification lacks explanation since it only provides a binary result without expressing which parts of a sentence could be the source of problem. The majority of the literature defines rumors as "an item of circulating information whose veracity status is yet to be verified at the time of posting" (Zubiaga et al., 2018). Providing explainability for unverified rumor detection is challenging. A related research area works with fake news is more well-studied since fake news has a verified veracity. Attribute information, linguistic features, and semantic meaning of post (Yang et al., 2019) and/or comments (Shu et al., 2019) have been used to provide explainability for fake news detection. A verified news database has to be established for these approaches. However, for rumor detection, sometimes a decision has to be made based on the current tweet only. Text-level models with explainability that recognize rumors by feature extraction should be developed to tackle this problem.

In this work, we propose LEX-GAN, a GAN-based layered explainable framework for text-level rumor detection. LEX-GAN keeps the ability of discriminating between real-world and generated samples, and also serves as a discriminator-oriented model that classifies real-world and generated fake samples. We overcome the infeasibility of propagating the gradient from discriminator back to the generator by applying Reinforcement Learning (RL) to train the layered generators. The training instability of long sentence generation is lowered by selectively replacing words in the sentence. We solve the per time step error attribution difficulty by word-level generation and evaluation. We show that our model outperforms the baselines in terms of addressing the degraded accuracy problem with clean samples only. The major contributions of this work are listed as follows:

- LEX-GAN delivers an explainable rumor detection without requiring a verified news database. Rumors could stay unverified for a long period of time because of information insufficiency. Providing explainability of which words in the sentence could be problematic is critical especially when there is no verified fact. When a verified news database is achievable, LEX-GAN can be applied to fake news detection with minor modification.

- The layered structure of LEX-GAN avoids the function mixture and boosts the performance. During framework design, we found that using one layer to realize two functions either in generative or discriminative model causes function mixture and hurts the performance. LEX-GAN generates high-quality rumors by first intelligently selecting words to be replaced, then choosing appropriate substitutes to replace. The explanation generation and rumor detection are realized separately by two layers in the discriminative model.

- LEX-GAN is a powerful framework in textural mutation detection. We demonstrate the mutation detection power by applying LEX-GAN to a gene classification and mutation detection task. LEX-GAN accurately identifies tokens in the gene sequences that are likely from the mutation, and classifies mutated gene sequences with high precision.

## 2 RELATED WORK

GANs in continuous space promise full differentiability. When dealing with discrete elements such as text, the discrete nature results in non-differentiability (Huszár, 2015). Researchers work with discrete element generation to either avoid the issue and reformulate the problem, or consider RL methods (Yu et al., 2017). Gulrajani et al. (Gulrajani et al., 2017) propose a fully differentiable Convolutional Neural Network (CNN)-based wasserstein GAN that produces sequences of 32 characters. However, the quality of generated sentences would be sufficient only if spelling errors are

frequent. RNN-based GANs, on the other hand, generate a sequence word by word, hence the quality issues such as spelling errors are rare. Many researchers have explored the non-differentiable problem in RNN-based GANs. Wolf et al. (Press et al., 2017) extend the work of (Gulrajani et al., 2017) by training RNN using curriculum learning. They start by training on short sequences and then slowly increase the sequence length. SeqGAN (Yu et al., 2017) bypasses the generator differentiation problem by performing gradient policy update. The RL reward signal coming from the discriminator is calculated on a complete sequence. Differently from SeqGAN, MaskGAN (Fedus et al., 2018) employs an actor-critic training procedure on a text infilling task designed to provide rewards at every time step.

Inspired by the thriving development of GANs on text applications, GAN-style rumor detection has become the state-of-the-art. GAN-GRU (Ma et al., 2019) is one of the latest GAN-based rumor detectors. By utilizing adversarial training, a generator is designed to create uncertainty. Hence, the complicated sequences pressurise the discriminator to learn stronger rumor representations. GAN-GRU reaches the highest accuracy of $78.1\%$ on a bench-marking rumor dataset PHEME (Kochkina et al., 2018). Although GAN-GRU outperforms other models, it does not offer explainability, nor does it provide any information regarding the parts of the sentences that are problematic. Addressing such deficiencies has been the main motivation for our work.

## 3 LEX-GAN

Figure 1 shows the architecture of our proposed LEX-GAN. In rumor detection task, the generators have to intelligently construct a rumor that appears like non-rumor to deceive the discriminators. Since a good lie usually has some truth in it, we choose to replace some of the tokens in the sequence and keep the majority to realize this goal. Two steps for intelligently replacing tokens in a sequence are: i) determine where to replace, and ii) choose what substitutes to use. $G_{where}$ and $G_{replace}$ are designed to realize these two steps. After constructing the strong generators, the discriminators are designed to provide a defense mechanism. Through adversarial training, the generators and discriminators grow stronger together, in terms of generating and detecting rumors, respectively. In rumor detection, given a sentence, there are two questions that need to be answered: i) is it a rumor or a non-rumor, and ii) if a rumor, which parts are problematic. $D_{classify}$ and $D_{explain}$ are designed to answer these two questions. We found that realizing two functions by one layer either in discriminative model or generative model hurts the performance. Hence the layered structure is designed.

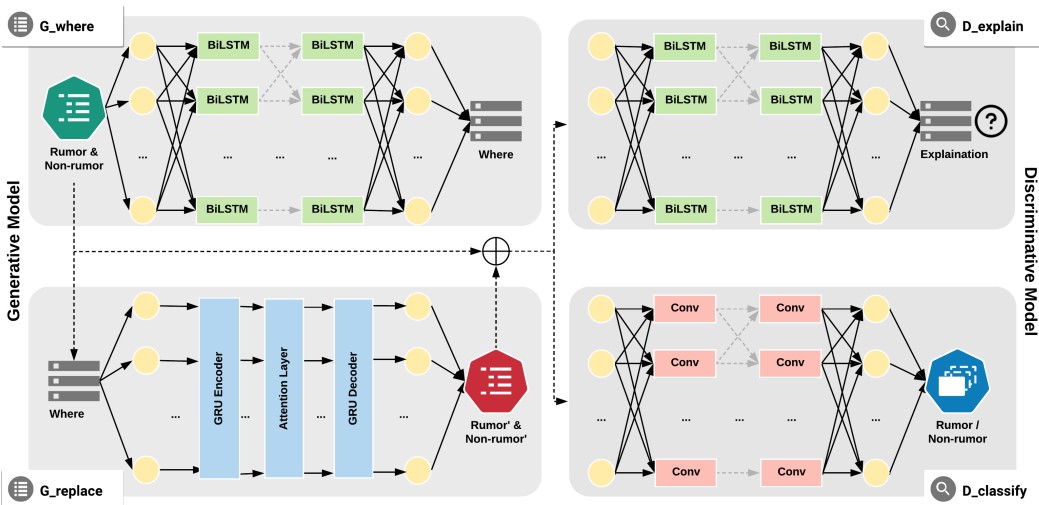

Figure 1: LEX-GAN framework. The generative model (shown on the left hand side) consists of two generators $G_{where}$ and $G_{replace}$. The discriminative model (shown on the right hand side) consists of two discriminators, namely $D_{explain}$ for explainability and $D_{classify}$ for classification.

## 3.1 GENERATIVE MODEL

The sequence generation task of LEX-GAN is done by the generative model: $G_{where}$ and $G_{replace}$. Given a human-generated real-world sequence input $\mathbf{x} = (x_1, x_2, ..., x_M)$ with length $M$, for example a tweet-level sentence containing $M$ words, $G_{where}$ outputs a probability vector $\mathbf{p} = (p_1, p_2, ..., p_M)$ indicating the probabilities of each item $x_i$ ($i \in [1, M]$) being replaced. $\mathbf{p}$ is applied to input $\mathbf{x}$ to construct a new sequence $\mathbf{x}^{where}$ with some items replaced by blank. For example, $x_2$ becomes a blank and $\mathbf{x}^{where} = (x_1, \_, ..., x_M)$.

$$\mathbf{x}^{where} = f(\mathbf{p}) \circ \mathbf{x} = f(G_{where}(\mathbf{x})) \circ \mathbf{x}, \tag{1}$$

where $f(\cdot)$ binarizes the input based on a hyperparameter $N_{replace}$. It determines the percentage of the words to be replaced in a sentence. Operator $\circ$ works as follows. If $a = 1$, then $a \circ b = b$. If $a = 0$, then $a \circ b = \_$. $G_{replace}$ is an encoder-decoder model with attention mechanism. It takes $\mathbf{x}^{where}$ and fills in the blank, then outputs a sequence $\mathbf{x}^{replace} = (x_1, x_2^{replace}, ..., x_M)$. The generative model is not fully differentiable because of the sampling operations on $G_{where}$ and $G_{replace}$. To train the generative model, we adopt policy gradients (Sutton et al., 2000) from RL to solve the non-differentiable issue.

### 3.1.1 $G_{replace}$ GRU-BASED ENCODER

Gated Recurrent Units (GRUs) (Cho et al., 2014) are improved versions of standard RNNs that use update gates and reset gates to resolve the vanishing gradient problem of a standard RNN. In our GRU-based encoder, the hidden state $h_t$ is computed as $GRU_{encoder}(x_t^{where}, h_{t-1})$:

$$h_t = (1 - z_t) \odot h_{t-1} + z_t \odot h_t', \tag{2}$$

$$z_t = \sigma(W_z^{enc} x_t^{where} + U_z^{enc} h_{t-1} + b_z^{enc}), \tag{3}$$

$$h_t' = tanh(W_h^{enc} x_t^{where} + U_h^{enc}(r_t \odot h_{t-1}) + b_h^{enc}), \tag{4}$$

$$r_t = \sigma(W_r^{enc} x_t^{where} + U_r^{enc} h_{t-1} + b_r^{enc}), \tag{5}$$

where $W_z^{enc}, W_h^{enc}, W_r^{enc}, U_z^{enc}, U_h^{enc}$ and $U_r^{enc}$ are encoder weight matrices. $\sigma(\cdot)$ is the sigmoid function. $\odot$ represents element-wise multiplication.

### 3.1.2 $G_{replace}$ GRU-BASED DECODER WITH ATTENTION MECHANISM

In LEX-GAN, our encoder-decoder $G_{replace}$ utilizes attention mechanism (Bahdanau et al., 2014) to automatically search for parts of a sentence that are relevant to predicting the target word. The content vector $c_t$ summarizes all the information of words in a sentence. It depends on the annotations $h_t$ and is computed as a weighted sum of these $h_t$:

$$c_t = \sum_{j=1}^{M} \alpha_{tj} h_j, \tag{6}$$

$$\alpha_{tj} = \frac{exp(e_{tj})}{\sum_{k=1}^{M} exp(e_{tk})}, \tag{7}$$

$$e_{tj} = a(s_{t-1}, h_j), \tag{8}$$

where $e_{tj}$ scores how well the inputs around position $j$ and the output at position $t$ match. Alignment model $a$ is a neural network that jointly trained with all other components. The GRU decoder takes the previous target $y_{t-1}$ and the context vector $c_t$ as input, and utilizes GRU to compute the hidden state $s_t$ as $GRU_{decoder}(y_{t-1}, s_{t-1}, c_t)$:

$$s_t = (1 - z_t') \odot s_{t-1} + z_t' \odot s_t', \tag{9}$$

$$z_t' = \sigma(W_z^{dec} y_{t-1} + U_z^{dec} s_{t-1} + C_z^{dec} c_t), \tag{10}$$

$$s_t' = tanh(W_s^{dec} y_{t-1} + U_s^{dec}(r_t' \odot s_{t-1}) + C_s^{dec} c_t), \tag{11}$$

$$r_t' = \sigma(W_r^{dec} y_{t-1} + U_r^{dec} s_{t-1} + C_r^{dec} c_t), \tag{12}$$

where $W_z^{dec}, W_s^{dec}, W_r^{dec}, U_z^{dec}, U_s^{dec}, U_r^{dec}, C_z^{dec}, C_s^{dec}$ and $C_r^{dec}$ are decoder weight matrices. Through this attention-equipped encoder-decoder, $G_{replace}$ intelligently replaces items in sequences and outputs adversarial samples.

## 3.2 DISCRIMINATIVE MODEL

The generated adversarial samples $\mathbf{x}^{replace}$ combined with original data $\mathbf{x}$ are fed to the discriminative model. $D_{classify}$ and $D_{explain}$ are trained independently. We note that the two discriminators can depend on each other, but we have chosen to explore the dependency as part of our future work on LEX-GAN. $D_{classify}$ provides a probability in rumor detection, and $D_{explain}$ provides the probability of each word in the sentence being problematic. The explainability of LEX-GAN is gained by adversarial training. We first insert adversarial items in the sequence, then train $D_{explain}$ to detect them. Through this technique, LEX-GAN can not only classify data with existing patterns, but also classify sequences with unseen patterns that may appear in the future. Adversarial training improves the robustness and generalization ability of LEX-GAN. We will show in Section 5 the effectiveness of LEX-GAN in addressing the accuracy reduction problem, when compared to prior work.

## 3.3 TRAINING

In rumor detection, a sequence $\mathbf{x}$ has a true label $Y$ being either a rumor $R$ or a non-rumor $N$. After manipulating the sequence $\mathbf{x}$, output of the generative model $\mathbf{x}^{replace}$ is labeled as $R$ since it is machine generated. The objective of a $\phi$-parameterized generative model is to mislead the $\theta$-parameterized discriminators. In our case, $D_{classify}^{\theta}(\mathbf{x}^{replace})$ indicates how likely the generated $\mathbf{x}^{replace}$ is classified as $N$. $D_{explain}^{\theta}(\mathbf{x}^{replace})$ indicates how accurately $D_{explain}^{\theta}$ detects the replaced words in a sequence. The error attribution per time step in LEX-GAN is achieved naturally since $D_{explain}^{\theta}$ evaluates each token and therefore provides a fine-grained supervision signal to the generators. For example, a case where the generative model produces a sequence that deceives the discriminative model. Then the reward signal from $D_{explain}^{\theta}$ indicates how well the position of each replaced word contributes to the error result. The reward signal from $D_{classify}^{\theta}$ represents how well the combination of the position and the replaced word deceived the discriminator. The generative model is updated by applying a policy gradient on the received rewards from the discriminative model.

The rumor generation problem is defined as follows. Given a sequence $\mathbf{x}$, $G_{where}^{\phi}$ is used to produce a sequence of probabilities $\mathbf{p}$ indicating the replacing probability of each token in $\mathbf{x}$. $G_{replace}^{\phi}$ takes $\mathbf{x}^{where}$ and produces a new sequence $\mathbf{x}^{replace}$. This newly generated $\mathbf{x}^{replace}$ is a sentence, part of which is replaced and labeled as $R$. At time step $t$, the state $\mathbf{s}$ consists of $\mathbf{s}^{where}$ and $\mathbf{s}^{replace}$. $\mathbf{s}^{where} = (p_1, ..., p_{t-1})$, $\mathbf{s}^{replace} = (x_1^{replace}, ..., x_{t-1}^{replace})$. The policy model $G_{where}^{\phi}(p_t|p_1, ..., p_{t-1})$ and $G_{replace}^{\phi}(x_t^{replace}|x_1^{replace}, ..., x_{t-1}^{replace})$ are stochastic. Following RL, $G_{where}^{\phi}$'s objective is to maximize its expected long-term reward:

$$J_{where}(\phi) = E[R_T|\mathbf{s}_0, \phi] = \sum_{p_1} G_{where}^{\phi}(p_1|\mathbf{s}_0^{where}) \cdot Q_{D^{\theta}}^{G^{\phi}}(\mathbf{s}_0^{replace}, \mathbf{a}), \tag{13}$$

$$Q_{D^{\theta}}^{G^{\phi}}(\mathbf{s}_0^{replace}, \mathbf{a}) = -D_{explain}^{\theta}(\mathbf{s}_0^{replace}) + D_{classify}^{\theta}(\mathbf{s}_0^{replace}), \tag{14}$$

where $Q_{D^{\theta}}^{G^{\phi}}(\mathbf{s}_0, \mathbf{a})$ is the accumulative reward following policy $G^{\phi}$ starting from state $\mathbf{s}_0 = \{\mathbf{s}_0^{where}, \mathbf{s}_0^{replace}\}$. $-D_{explain}^{\theta}(\mathbf{s}^{replace})$ indicates how much the generative model misleads $D_{explain}^{\theta}$. $\mathbf{a}$ is an action set that contains output of both $G_{where}^{\phi}$ and $G_{replace}^{\phi}$. $R_T$ is the reward for a complete sequence. Similarly to $G_{where}^{\phi}$, $G_{replace}^{\phi}$ maximizes its expected long-term reward:

$$J_{replace}(\phi) = \sum_{x_1^{replace}} G_{replace}^{\phi}(x_1^{replace}|\mathbf{s}_0^{replace}) \cdot Q_{D^{\theta}}^{G^{\phi}}(\mathbf{s}_0^{replace}, \mathbf{a}). \tag{15}$$

We apply a discriminative model provided reward value to the generative model after the sequence is produced. The reason is that our $G_{replace}^{\phi}$ doesn't need to generate each and every word in the sequence, but only fills a few blanks that are generated by $G_{where}^{\phi}$. Under this assumption, long-term reward is approximated by the reward gained after the whole sequence is finished.

The discriminative model and the generative model are updated alternately. The loss function of discriminative model is defined as follows:

$$L_D = \lambda_D^{explain} L_D^{explain} + \lambda_D^{classify} L_D^{classify}, \tag{16}$$

$$L_D^{explain} = -E_{y \sim f(G_{where}^{\phi}(\mathbf{x}))}[ylog(D_{explain}^{\theta}(\mathbf{x}^{replace})) + (1-y)log(1 - D_{explain}^{\theta}(\mathbf{x}^{replace}))] \quad (17)$$

$$L_D^{classify} = -E_{y \sim Y}[ylog(D_{classify}^{\theta}(\mathbf{x}^{replace})) + (1-y)log(1 - D_{classify}^{\theta}(\mathbf{x}^{replace}))] \quad (18)$$

where $\lambda_D^{explain}$ and $\lambda_D^{classify}$ are the balancing parameters.

We adopt the training method in GANs to train the networks. In each epoch, the generative model and the discriminative model are updated alternately. Over-training the discriminators or the generators may result in a training failure. Thus hyper-parameters $G_{STEP}$ and $D_{STEP}$ are introduced to balance the training. In each epoch, the generators are trained $G_{STEP}$ times. Then discriminators are trained $D_{STEP}$ times.

## 4 EXPERIMENT SETTINGS

### 4.1 DATASETS

We evaluate LEX-GAN on a benchmark Twitter rumor detection dataset PHEME (Kochkina et al., 2018), and a splice site benchmark dataset NN269 (Reese et al., 1997). PHEME has two versions. PHEMEv5 contains 5792 tweets related to five news, 1972 of them are rumors and 3820 of them are non-rumors. PHEMEv9 contains 6411 tweets related to nine news, 2388 of them are rumors and 4023 of them are non-rumors. NN269 dataset contains 13231 splice site sequences. It has 6985 acceptor splice site sequences with length of 90 nucleotides, 5643 of them are positive $AP$ and 1324 of them are negative $AN$. It also has 6246 donor splice site sequences with length of 15 nucleotides, 4922 of them are positive $DP$ and 1324 of them are negative $DN$.

### 4.2 MODELS

In the rumor detection task, we compare LEX-GAN with six popular rumor detectors: RNN with LSTM cells, CNN, VAE-LSTM, VAE-CNN, a contextual embedding model with data augmenting (DATA-AUG) (Han et al., 2019), and a GAN-based rumor detector (GAN-GRU) (Ma et al., 2019). One of the strengths of LEX-GAN is that under the delicate layered structure that we designed, the choice of model structure effects the results but not significantly. To showcase this ability of the layered structure, we generate a variation of LEX-GAN as one baseline. LEX-LSTM is generated by replacing LEX-GAN's $G_{replace}$ with a LSTM model. LEX-GAN generates a set of sequences by substituting around 10% of the words in original sequences. We pre-train the $D_{classify}$ by fixing $N_{replace} = 10\%$. We then freeze $D_{classify}$ and train the other three models. During training, we lower $N_{replace}$ from 50% to 10% to guarantee data balancing for $D_{explain}$ and hence better results in terms of explainability. All the embedding layers in the generators and discriminators are initialized with 50 dimension GloVe (Pennington et al., 2014) pre-trained vectors. Early stopping technique is applied during training. LEX-LSTM is trained under the same training process. The generated data in the rumor task are labeled as $R$, and we denote this dataset as PHEME'. For fairness and consistency, we train baselines LSTM, CNN, VAE-LSTM, and VAE-CNN with PHEME and PHEME+PHEME'. For all baselines, we use two evaluation principles: (i) hold out 10% of the data for model tuning. (ii) Leave-one-out (L) principle, i.e., leave out one news for test, and train the models on other news. Final results are calculated as the weighted average of all results. L principle constructs a realistic testing scenario and evaluates the rumor detection ability under new out-of-domain data. For DATA-AUG and GAN-GRU, we import the best results reported in their papers.

For the gene classification and mutation detection task, LEX-GAN generates a dataset NN269' by replacing nine characters in acceptor sequences and three characters in donor sequences. We compare LEX-GAN with six models: RNN with LSTM cells, CNN, VAE-LSTM, VAE-CNN, LEX-LSTM, and a state-of-the-art splice site predictor EFFECT from (Kamath et al., 2014). The first four baselines are trained under NN269+NN269', and tested on both NN269+NN269' and clean data NN269. We import EFFECT's results from (Kamath et al., 2014). To evaluate LEX-GAN's generalization ability, we label the generated sequences by the following rule: if the input sequence $\mathbf{x}$ has label $Y$, then the output sequence $\mathbf{x}^{replace}$ is labeled as $Y'$, indicating that $\mathbf{x}^{replace}$ is from class $Y$ but with modification. The final classification output of LEX-GAN is two-fold: $AP$, $AN$ for acceptor, or $DP$, $DN$ for donor. We merge the generated classes $AP'$, $AN'$ and $DP'$, $DN'$

with original classes to evaluate the generalization ability of LEX-GAN. Given a sequence, LEX-GAN can classify it into one of the known classes, although the sequence could be either clean or modified.

Architecture setup of LEX-GAN is as follows. $G_{where}$ is a RNN with Bidirectional LSTM (BiLSTM). It has two hidden BiLSTM layers followed by a dense layer. The one we used in all experiments has the architecture of EM-32-32-16-OUT, where EM, OUT represent embedding and output, respectively. $G_{replace}$ is an encoder-decoder with attention mechanism. The encoder has two GRU layers, and the decoder has two GRU layers equipped with attention mechanism. The architecture of $G_{replace}$ we used in all experiments is EM-64-64-EM-64-64-OUT. $D_{explain}$ has the same architecture as $G_{where}$. $D_{classify}$ is a CNN with two convolutional layers followed by a dense layer. The one we used in all experiments has the architecture of EM-32-64-16-OUT. LEX-LSTM utilizes an LSTM-based encoder-decoder with architecture EM-32-32-EM-32-32-OUT as $G_{replace}$. For baselines architecture details, see Appendix A.

Table 1: Macro-f1 and accuracy comparison between LEX-GAN and baselines on the rumor detection task. BOTH represents the models are trained on PHEME+PHEME'. * indicates the best result from the work that proposed the corresponding model. L represents the model is evaluated under leave-one-out principle.

| | | PHEMEv5 | | | | PHEMEv9 | | | |
| | | PHEME | | PHEME+PHEME' | | PHEME | | PHEME+PHEME' | |
| | | Macro-f1 | Accuracy | Macro-f1 | Accuracy | Macro-f1 | Accuracy | Macro-f1 | Accuracy |
|---|---|---|---|---|---|---|---|---|---|
| LSTM | PHEME | 0.6425 | 0.6542 | 0.4344 | 0.4345 | 0.6261 | 0.6269 | 0.4999 | 0.5283 |
| | BOTH | 0.5413 | 0.5761 | 0.6421 | 0.6621 | 0.4938 | 0.5341 | 0.5937 | 0.6103 |
| CNN | PHEME | 0.6608 | 0.6660 | 0.4792 | 0.4833 | 0.6549 | 0.6552 | 0.5028 | 0.5253 |
| | BOTH | 0.4936 | 0.5273 | 0.5757 | 0.5927 | 0.5025 | 0.5400 | 0.5880 | 0.6040 |
| VAE-LSTM | PHEME | 0.4677 | 0.5625 | 0.2582 | 0.2871 | 0.4454 | 0.4589 | 0.4231 | 0.4326 |
| | BOTH | 0.4587 | 0.4589 | 0.5001 | 0.5595 | 0.3349 | 0.4960 | 0.2148 | 0.2592 |
| VAE-CNN | PHEME | 0.5605 | 0.5605 | 0.4655 | 0.4902 | 0.3859 | 0.5029 | 0.2513 | 0.2778 |
| | BOTH | 0.5395 | 0.5429 | 0.4838 | 0.5302 | 0.5164 | 0.5166 | 0.4573 | 0.4794 |
| GAN-GRU | PHEME | 0.7810* | 0.7810* | - | - | - | - | - | - |
| LEX-LSTM | PHEME | 0.8242 | 0.8242 | 0.6259 | 0.6302 | 0.8066 | 0.8066 | 0.6884 | 0.7044 |
| LEX-GAN | PHEME | **0.8475** | **0.8476** | **0.6524** | **0.6777** | **0.8084** | **0.8095** | **0.7620** | **0.8085** |
| LSTM (L) | PHEME | 0.5693 | 0.6030 | 0.5260 | 0.5710 | 0.5217 | 0.5827 | 0.5055 | 0.5906 |
| | BOTH | 0.3854 | 0.4478 | 0.4980 | 0.6572 | 0.4120 | 0.4933 | 0.5050 | **0.6729** |
| CNN (L) | PHEME | 0.5994 | 0.6406 | 0.5324 | 0.5779 | 0.5477 | 0.6035 | 0.5051 | 0.5769 |
| | BOTH | 0.4265 | 0.4719 | 0.5256 | **0.6587** | 0.3679 | 0.4601 | 0.4562 | 0.6455 |
| VAE-LSTM (L) | PHEME | 0.3655 | 0.3996 | 0.3620 | 0.3959 | 0.4256 | 0.5367 | 0.4284 | 0.5397 |
| | BOTH | 0.3919 | 0.5198 | 0.3876 | 0.5174 | 0.4225 | 0.5442 | 0.4270 | 0.5442 |
| VAE-CNN (L) | PHEME | 0.4807 | 0.5190 | 0.4816 | 0.5214 | 0.4316 | 0.4597 | 0.4314 | 0.4587 |
| | BOTH | 0.4594 | 0.5320 | 0.4662 | 0.5380 | 0.4686 | 0.5347 | 0.4786 | 0.5411 |
| DATA-AUG (L) | PHEME | 0.5350* | **0.7070*** | - | - | - | - | - | - |
| LEX-LSTM (L) | PHEME | 0.6666 | 0.6866 | 0.5703 | 0.6411 | 0.5972 | 0.6272 | 0.5922 | 0.6371 |
| LEX-GAN (L) | PHEME | **0.6745** | 0.7016 | **0.6126** | 0.6342 | **0.6207** | **0.6438** | **0.6016** | 0.6400 |

# 5 RESULTS

## 5.1 RUMOR DETECTION

Comparison between LEX-GAN $D_{classify}$ and baselines in the rumor detection task is shown in Table 1. In this experiment, we use PHEME data to train LEX-GAN. During training, LEX-GAN generates PHEME' to enhance the discriminative model. In real world applications, original clean dataset is available all the time. However, the modified or adversarial data that contains different patterns are not always accessible. Models like LSTM and CNN do not have generalization ability and usually perform worse facing adversarial input. Generative models such as GANs are more robust. In VAE-LSTM and VAE-CNN, we first pre-train VAEs, then LSTM and CNN are trained under latent representations of pre-trained VAEs. Under the first evaluation principle (described in Section 4.2), LEX-GAN and LEX-LSTM outperform all baselines in terms of both macro-f1 and accuracy. Accuracy is not sufficient when the test data are not balanced, hence macro-f1 is provided for comprehensive comparison. Under the first evaluation principle, the robustness and generalization ability of LEX-GAN and LEX-LSTM are shown by comparing with baselines under PHEME+PHEME'. LEX-GAN reaches the highest values in both versions of PHEME+PHEME' and LEX-LSTM follows as the second best. Under L principle, LEX-GAN and LEX-LSTM achieves highest macro-f1

scores in all cases. These results confirm the rumor detection ability of the proposed layered structure under new, out-of-domain data. Adversarial training of baselines improves generalization and robustness under PHEME+PHEME', but hurts the performance under clean data as expected. Although LEX-GAN and LEX-LSTM are trained adversarially, they achieve the highest macro-f1 under clean data PHEME. The results confirm that LEX-GAN outperforms the baselines in terms of addressing accuracy reduction problem.

LEX-GAN's $D_{explain}$ recognizes the modified parts in sequences accurately. Its macro-f1 on PHEME'v5 and PHEME'v9 are $80.42\%$ and $81.23\%$, respectively. Examples of $D_{explain}$ predicting suspicious parts in rumors are shown in Table 2. In the first rumor, "hostage escape" is the most important part in the sentence, and if these two words are problematic, then the sentence is highly likely to be problematic. Given an unverified or even unverifiable rumor, $D_{explain}$ provides reasonable explanation without requiring a previously collected verified news database. Additional results of LEX-GAN's comparison with baselines can be found in Appendix B.

Table 2: Examples of $D_{explain}$ predicting suspicious words in rumors (marked in bold). $D_{classify}$ outputs probabilities in range $[0, 1]$, where 0 and 1 represents rumor and non-rumor, respectively.

| | |
|---|---|
| 0.0010 | breaking update 2 **hostages escape** lindt café through front **door** 1 via fire door url sydneysiege url |
| 0.0255 | **newest** putin **rumour** his girlfriend just gave birth to their child url cdnpoli russia |
| 0.0300 | soldier **gets cpr** after being shot at war memorial in ottawa url |
| 0.0465 | sydney's **central business district** is under lockdown as gunman takes hostages at a cafe live **stream** as it unfolds url |
| 0.2927 | so in **5mins** mike brown **shaved** his head and changed his **scandals** to **shoes** i think your being lied too ferguson url |

## 5.2 GENE CLASSIFICATION AND MUTATION DETECTION

In this experiment, all models are trained under NN269+NN269' to ensure fairness. When test with NN269+NN269', there are 8 classes in total: $AP$, $AN$, $DP$, $DN$ from NN269 and $AP'$, $AN'$, $DP'$, $DN'$ from NN269'. If solely clean data from NN269 is accessible during training, then LEX-GAN and LEX-LSTM are the only models that can recognize if a given sequence is modified or unmodified. Comparison between LEX-GAN's (and LEX-LSTM's) $D_{classify}$ and baselines is shown in Table 3. Under long acceptor data, baselines perform significantly worse than LEX-GAN and LEX-LSTM. Under short donor data, LEX-GAN and LEX-LSTM achieve highest AURoCs. This implies that LEX-GAN and LEX-LSTM are stronger when the input are long sequences. The layered structure and adversarial training under the augmented dataset provide LEX-GAN the ability of extracting meaningful patterns from long sequences. For short sequences, LEX-GAN and LEX-LSTM provide highest AURoC, and simpler models such as CNN can also give good classification results. This is because for short sequences, textural feature mining and understating is relatively easier then in long sequence. Under NN269', LEX-GAN's $D_{classify}$ and $D_{explain}$ achieve $92.25\%$ and $72.69\%$ macro-f1, respectively. Additional gene mutation detection experiments can be found in Appendix B.2.

## 5.3 LIMITATIONS AND ERROR CASES IN RUMOR DETECTION

Examples of error cases of LEX-GAN in rumor detection task are presented in Table 4. For some short sentences, $D_{explain}$ sometimes fails to predict the suspicious parts. The reason is that the majority of LEX-GAN's training data are long sentences, hence for short sentences, $D_{explain}$ does not perform as well as in long sentences. This problem could be solved by feeding more short sentences to LEX-GAN. In most cases, although $D_{explain}$ does not generate predictions, $D_{classify}$ still can provide accurate classification. As shown in Table 4, $D_{classify}$ outputs low score, i.e., classifies the input as rumor, for four out of five rumors.

## 6 DISCUSSION

In this work, we proposed LEX-GAN, a layered explainable text-level rumor detector based on GAN. We used the policy gradient method to effectively train the layered generators. LEX-GAN

Table 3: Comparison between LEX-GAN and baselines on the gene classification and mutation detection task. * indicates the best result from the corresponding paper. 2-class refers to $AP$, $AN$ for acceptor, and $DP$, $DN$ for donor. 4-class refers to $AP$, $AN$, $AP'$, $AN'$ for acceptor, and $DP$, $DN$, $DP'$, $DN'$ for donor. A and D indicate acceptor and donor.

| | NN269 (2-class) | | | NN269+NN269' (2-class) | | | NN269+NN269' (4-class) | | |
|---|---|---|---|---|---|---|---|---|---|
| | Macro-f1 | Accuracy | AURoC | Macro-f1 | Accuracy | AURoC | Macro-f1 | Accuracy | AURoC |
| LSTM (A) | 0.8120 | 0.8870 | 0.9305 | 0.7794 | 0.8580 | 0.9036 | 0.7800 | 0.8580 | 0.9715 |
| CNN (A) | 0.5663 | 0.7933 | 0.6324 | 0.5594 | 0.7808 | 0.6131 | 0.5593 | 0.7808 | 0.8875 |
| VAE-LSTM (A) | 0.7664 | 0.8566 | 0.8451 | 0.6781 | 0.8323 | 0.7780 | 0.6531 | 0.8342 | 0.8806 |
| VAE-CNN (A) | 0.5657 | 0.7539 | 0.6135 | 0.5744 | 0.7651 | 0.6219 | 0.5379 | 0.7470 | 0.8411 |
| EFFECT (A) | - | - | 0.9770* | - | - | - | - | - | - |
| LEX-LSTM (A) | 0.9131 | 0.9458 | 0.9781 | 0.8794 | 0.9243 | 0.9658 | 0.8758 | 0.9223 | 0.9879 |
| LEX-GAN (A) | **0.9175** | **0.9494** | **0.9807** | **0.8831** | **0.9301** | **0.9691** | **0.8839** | **0.9311** | **0.9894** |
| LSTM (D) | 0.8336 | 0.8214 | 0.9003 | 0.8148 | 0.7998 | 0.8802 | 0.7648 | 0.7530 | 0.9246 |
| CNN (D) | 0.9131 | 0.9393 | 0.9795 | **0.9025** | **0.9323** | 0.9746 | **0.8336** | **0.8583** | 0.9596 |
| VAE-LSTM (D) | 0.8011 | 0.8515 | 0.9218 | 0.7336 | 0.8329 | 0.8217 | 0.5774 | 0.7692 | 0.9194 |
| VAE-CNN (D) | 0.8386 | 0.8772 | 0.9554 | 0.7909 | 0.8593 | 0.8528 | 0.5585 | 0.7415 | 0.9190 |
| EFFECT (D) | - | - | 0.9820* | - | - | - | - | - | - |
| LEX-LSTM (D) | 0.9272 | 0.9484 | **0.9822** | 0.8802 | 0.9140 | **0.9766** | 0.8113 | 0.8580 | 0.9541 |
| LEX-GAN (D) | **0.9274** | **0.9494** | 0.9810 | 0.8988 | 0.9296 | 0.9635 | 0.8119 | 0.8470 | **0.9776** |

Table 4: Examples of $D_{explain}$ failing to predict suspicious words in some short rumors. $D_{classify}$ outputs probabilities in range $[0, 1]$, where 0 and 1 represents rumor and non-rumor, respectively.

| | |
|---|---|
| 0.0112 | ottawa police report a third shooting at rideau centre no reports of injuries |
| 0.0118 | breaking swiss art museum accepts artworks bequeathed by late art dealer gurlitt url |
| 0.0361 | breaking germanwings co pilot was muslim convert url |
| 0.5771 | the woman injured last night ferguson url |
| 0.4451 | germanwings passenger plane crashes in france url |

outperforms the baseline models in mitigating the accuracy reduction problem, that exists in case of only clean data. We demonstrate the classification ability and generalization power of LEX-GAN by applying it to two applications: rumor detection and gene classification and mutation detection.

On average, in the rumor detection task, LEX-GAN outperforms the baselines on clean dataset PHEME and enhanced dataset PHEME+PHEME' by $26.85\%$ and $17.04\%$ in terms of macro-f1, respectively. LEX-GAN provides reasonable explanation without a previously constructed verified news database, and achieves significantly high performance. In the gene classification and mutation detection task, LEX-GAN identifies the mutated gene sequence with high precision. On average, LEX-GAN outperforms baselines in both NN269 and NN269+NN269' (2-class) by $10.71\%$ and $16.06\%$ in terms of AURoC, respectively. In both rumor detection and gene mutation detection tasks, LEX-GAN's explainability is demonstrated by identifying the mutations accurately (above $70\%$ macro-f1). We find that using two discriminators to perform classification and explanation separately achieves higher performance than using one discriminator to realize both functions. We also found the pre-train of $D_{classify}$ and varying $N_{replace}$ contribute to the high accuracy of $D_{explain}$. As part of our future work, we would like to explore the application of hierarchical attention network (Yang et al., 2016) to improve the performance of our generative model. We will also investigate the dependencies between the discriminators of LEX-GAN.

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

## A  Baselines Architecture Setup

The architectures of baselines LSTM, CNN, VAE-LSTM, and VAE-CNN used in both tasks are defined as in Table 5. VAE-LSTM and VAE-CNN use a pre-trained VAE followed by LSTM and CNN with the architectures we defined in Table 5. The VAE we pre-trained is a LSTM-based encoder-decoder. The encoder with architecture EM-32-32-32-OUT has two LSTM layers followed by a dense layer. The decoder has the architecture IN-32-32-OUT, where IN stands for input layer.

Table 5: Baselines' architecture.

| Model | Gene mutation detection task | Rumor detection task |
|---|---|---|
| LSTM | EM-LSTM(64)-LSTM(32)-DENSE(8)-OUT | EM-LSTM(32)-LSTM(16)-DENSE(8)-OUT |
| CNN | EM-CONV(32)-CONV(64)-DENSE(16)-OUT | EM-CONV(32)-CONV(16)-DENSE(8)-OUT |
| VAE-LSTM | LSTM(32)-LSTM(32)-DENSE(8)-OUT | LSTM(32)-LSTM(16)-DENSE(8)-OUT |
| VAE-CNN | CONV(32)-CONV(64)-DENSE(16)-OUT | CONV(32)-CONV(64)-DENSE(16)-OUT |

## B  Additional Samples

### B.1  Rumor Detection Task

Additional results for the rumor detection task are shown in Table 6. $D_{explain}$ provides reasonable explanation in most rumors. Here we also provide two examples to demonstrate the rumor detection

Table 6: Examples of $D_{explain}$'s prediction on rumors.

| |
| --- |
| **twin** hostage **situations erupt** in paris two victims have been killed our updated story url |
| pilots of crashed germanwings flight declared emergency at 10 47 a m as plane **fell rapidly** url |
| **doomed** germanwings co pilot **'suffered burnout** or depression' years before crash url url |
| sky news australia a **sixth** hostage has escaped from the lindt cafe in sydney sydneysiege |
| **approximately 50** hostages may be held **captive** at lindt café – local reports url sydneysiege url |

Table 7: Examples of $D_{explain}$ and $D_{classify}$'s prediction on rumor (first) and non-rumor (second). The suspicious words in the rumor predicted by $D_{explain}$ are marked in bold. $D_{classify}$ provides a score ranges from 0 to 1. 0 and 1 represent rumor and non-rumor, respectively.

| | |
| --- | --- |
| 0.1579 | **who's** your pick for worst **contribution** to sydneysiege **mamamia uber** or the daily tele |
| 0.8558 | glad to hear the sydneysiege is over but saddened that it even happened to begin with my heart goes out to all those affected |

power of LEX-GAN compared to baselines. Two examples that are correctly detected by LEX-GAN but incorrectly detected by other baselines is shown in Table 7.

For the first rumor, baselines CNN, LSTM, VAE-CNN, and VAE-LSTM provide scores 0.9802, 0.9863, 0.4917, and 0.5138, respectively. LEX-GAN provides a very low score for a rumor, while other baselines all generated relatively high scores, and even detect it as non-rumor. This is a very difficult example since from the sentence itself, we as human rumor detection agents even cannot pick the suspicious parts confidently. However, LEX-GAN gives a reasonable prediction and shows that it has the ability to understand and analyze complicated rumors. For the second non-rumor, baselines CNN, LSTM, VAE-CNN, and VAE-LSTM provide scores 0.0029, 0.1316, 0.6150, and 0.4768, respectively. In this case, a non-rumor sentence gains a high score from LEX-GAN, but several relatively low scores from the baselines. This example again confirms that our proposed LEX-GAN indeed captures the complicated nature of rumors and non-rumors.

### B.2 GENE CLASSIFICATION AND MUTATION DETECTION TASK ON GENE DATASET

We evaluate LEX-GAN on a molecular biology splice-junction gene sequences dataset (GENE) (Dua & Graff, 2017). GENE dataset contains 3109 splice junction gene sequences, and each sequence has 60 characters. Splice junctions are points on a DNA sequence at which superfluous DNA is removed during the process of protein creation in higher organisms (Dua & Graff, 2017). They are labeled as exon/intron boundaries ($EI$ sites), intron/exon boundaries ($IE$ sites), or neither ($N$).

For the gene classification and mutation detection task, LEX-GAN generates a dataset GENE' by replacing five characters in each sequence. We compare LEX-GAN with five models: RNN with LSTM cells, CNN, VAE-LSTM, VAE-CNN, and a DNA classifier (DNAC) from (Deshpande & Karypis, 2002). The first four baselines are trained under GENE+GENE', and tested on both GENE+GENE' and clean data GENE. We import DNAC's results from (Deshpande & Karypis, 2002). To evaluate LEX-GAN's generalization ability, we label the generated sequences by the following rule: if the input sequence $\mathbf{x}$ has label $Y$, then the output sequence $\mathbf{x}^{replace}$ is labeled as $Y'$, indicating that $\mathbf{x}^{replace}$ is from class $Y$ but with modification. The final classification output of LEX-GAN is three-fold: $EI$, $IE$, or $N$. We merge the generated classes $EI'$, $IE'$, $N'$ with original classes to evaluate the generalization ability of LEX-GAN. Given a sequence, LEX-GAN can classify it into one of the three known classes, although the sequence could be either clean or modified.

In this experiment, all models are trained under GENE+GENE' to ensure fairness. When test with GENE+GENE', there are 6 classes in total: $EI$, $IE$, $N$ from GENE and $EI'$, $IE'$, $N'$ from GENE'. If solely clean data from GENE are accessible during training, then LEX-GAN is the only model that can recognize if a given sequence is modified or unmodified. LEX-GAN can endue the baselines with this ability by generating GENE'. Comparison between LEX-GAN's $D_{classify}$ and baselines is shown in Table 8. Baselines perform significantly worse than LEX-GAN in 6-class classification.

This implies that the baselines can only roughly classify a modified sequence into a known class as shown in 3-class experiment, but their fine-grained classification ability is significantly lower than LEX-GAN. On GENE', $D_{classify}$ and $D_{explain}$ achieve $91.80\%$ and $89.49\%$ macro-f1, respectively. An example of $D_{explain}$'s prediction is shown in Table 9. This suggests that LEX-GAN can not only identify whether a gene-sequence is modified from which known class, but also provide an accurate prediction that explains which part of the sequence is modified.

Table 8: Comparison between LEX-GAN and baselines on the gene classification and mutation detection task. * indicates the best result from the corresponding paper. Note that DNAC does binary classification between $EI$ and $IE$, hence its result is relatively higher than 3-class classification. 3-class refers to $EI$, $IE$, $N$. 6-class refers to $EI$, $IE$, $N$, $EI'$, $IE'$, $N'$.

| | GENE (3-class) | | | GENE+GENE' (3-class) | | | GENE+GENE' (6-class) | | |
|---|---|---|---|---|---|---|---|---|---|
| | Precision | Recall | Macro-f1 | Precision | Recall | Macro-f1 | Precision | Recall | Macro-f1 |
| LSTM | 0.7894 | 0.7930 | 0.7911 | 0.7971 | 0.7978 | 0.7972 | 0.7931 | 0.7990 | 0.7953 |
| CNN | 0.7233 | 0.7306 | 0.7265 | 0.7270 | 0.7301 | 0.7280 | 0.7146 | 0.7207 | 0.7165 |
| VAE-LSTM | 0.4733 | 0.3835 | 0.3494 | 0.5184 | 0.4899 | 0.4806 | 0.2851 | 0.3583 | 0.3086 |
| VAE-CNN | 0.8360 | 0.8583 | 0.8441 | 0.8210 | 0.8321 | 0.8236 | 0.3571 | 0.4605 | 0.3870 |
| DNAC | - | - | 0.9390* | - | - | - | - | - | - |
| LEX-GAN | **0.9510** | **0.9642** | **0.9571** | **0.9391** | **0.9418** | **0.9403** | **0.9266** | **0.9336** | **0.9297** |

Table 9: Examples of the generative model modifies gene sequences and the discriminative model detects the modifications (marked in bold).

| | |
|---|---|
| Original | GGGCCCTGGCCCTGACCCAGACCTGGGCGCGTGAGTGCAGGGTCTGCAGGGAAATGGTCG |
| Modified | GGGCCCTGGCCCTGACCCAGACCTGGGCGCGTGAGTGCAGGGTCTGCAGGGAAATGG**TAS** |
| Prediction | GGGCCCTGGCCCTGACCCAGACCTGGGCGCGTGAGTGCAGGGTCTGCAGGGAAATGG**TAS** |
| Original | CCCAGGAGGGGTGGACCCACAGCCCAGGGAGGCCGAAAGCGCGGGCGGGCAGGCAGAGGC |
| Modified | **AC**CAGGAGGGGTGGACCCAC**N**GCCCAGGGAGGCCGAAAGCGCGGGCGGGCAGGCAGAGGC |
| Prediction | **AC**CAGGAGGGGTGGACCCAC**N**GCCCAGGGAGGCCGAAAGCGCGGGCGGGCAGGCAGAGGC |
| Original | TAATCGTTGATTCCCTTCCCTCCCTCACAGAAAGCATCCCTGGAGAACAGCCTGGAGGAG |
| Modified | TAATCGT**GGSGDT**CCCTTCCCTCCCTCACAGAAAGCATCCCTGGAGAACAGCCTGGAGGAG |
| Predicted | TAATCGTGG**SGDT**CCCTTCCCTCCCTCACAGAAAGCATCCCTGGAGAACAGCCTGGAGGAG |
| Original | CCCAGGAGGGGTGGACCCACAGCCCAGGGAGGCCGAAAGCGCGGGCGGGCAGGCAGAGGC |
| Modified | **AC**CAGGAGGGGTGGACCCAC**G**GCCCAGGGAGGCCGAAAGCGCGGGCGGGCAGGCAGAGGC |
| Prediction | **AC**CAGGAGGGGTGGACCCAC**G**GCCCAGGGAGGCCGAAAGCGCGGGCGGGCAGGCAGAGGC |

