# OpenReview forum: "LEX-GAN: Layered Explainable Rumor Detector Based on Generative Adversarial Networks"
_ICLR.cc/2020/Conference — Reject_

### Official Review · AnonReviewer1 · 2019-10-16
**Official Blind Review #1**

**Rating:** 3

**Review:**

In the paper, authors proposed a generative adversarial network-based rumor detection model that can label short text like Twitter posts as rumor or not. The model can further highlight the words that are responsible for the rumor accusation.

Proposed model consists of 4 sub models: a G_Where model finds the word to replace so to create an artificial rumor; a G_replace model decides what the replacement word should be; a D_classify model detects if a sequence is a rumor; a final D_explain model pinpoints the word of concern. D_ models and G_ models are trained in an adversarial competing way.

Experiments showed that the LEX-GAN model outperforms other non-GAN models by a large margin on a previously published rumor dataset (PHEME) and in a gene classification task.

My questions:

1) The task modeled is essentially a word replacement detection problem. Is this equivalent to rumor detection? Even if it performs really well on a static dataset, it could be very vulnerable to attackers. Various previous works mentioned in the paper, including the PHEME paper by Kochkina et al, used supporting evidence for detection, which sounds like a more robust approach.

2) Authors didn't explain the rationale behind the choice of model structure, e.g. GRU vs LSTM vs Conv. The different structures have been used in mix in the paper. Are those choices irrelevant or critical?

3) I would like to see more discussion on the nature of errors from those models, but it's lacking in the paper. This could be critical to understand the model’s ability and limitation, esp given that it’s not looking at supporting evidences from other sequences.

Small errors noticed: The citation for PHEME paper (Kochkina et al) points to a preprint version, while an ACL Anthology published version exists.

**Experience Assessment:**

I have read many papers in this area.

**Review Assessment: Checking Correctness Of Derivations And Theory:**

N/A

**Review Assessment: Checking Correctness Of Experiments:**

I assessed the sensibility of the experiments.

**Review Assessment: Thoroughness In Paper Reading:**

I read the paper at least twice and used my best judgement in assessing the paper.

---

> ### Author Response · Authors · 2019-11-11
> **Response to Review #1**
>
> We are very thankful to the reviewer for the comments.
> 1.	We use the word replacement to augment the available dataset and use both the generated data and original data to train the model. Therefore, the model is essentially trained to recognize rumor and detect detailed glitches. Compared to other works, LEX-GAN uses GAN and essentially adversarial training techniques are utilized as well, hence it is more robust to attacks. Kochkina’s work includes rumor detection and stance classification. The support evidence you pointed out is used in stance classification. In stance classification, posts and comments are labeled as support, deny, comment, or query due to their orientation toward rumor’s veracity. Stance classification and rumor detection are two different components in rumor classification system, which include four steps: rumor detection, tracking, stance classification, and veracity classification. LEX-GAN realizes rumor detection task based on the assumption that the rumor detection is done in the early stage when its veracity is unverified, and the comments are unavailable.
> 2.	We thank the reviewer for this important suggestion. To explain the model choice, we start with human thinking process while reading text. When we read a sentence, we don’t think from scratch, instead, we understand words based on previous words. This is the fundamental reason why we choose RNN models over ordinary neural network. LSTM and GRU are two commonly used RNN models in text data processing. Gwhere chooses which words in a sentence to be replaced and it has to consider the past words and the whole sentence, hence we choose LSTM to realize this function. The choice of Dexplain follows a similar reasoning. Greplace does the actual word-replacing work, the choice of GRU considers both performance and efficiency. GRU is computationally more efficient than LSTM and provides better results when used in Greplace in this work. CNN is also frequently applied in nantural language processing applications to do classification. Dclassify utilizes a CNN to realize a classification between rumor and non-rumor. In summary, RNN models and CNN models are not mutual exclusive under text data, the choice of a hybrid model structure follows the consideration of both performance and efficiency. One of the strengths of LEX-GAN is that under the delicate layered structure that we designed, the choice of model structure effects the results but not significantly, hence LEX-GAN can by deployed into a broad range of applications while maintaining a high level of performance. We generate a variation of LEX-GAN as a baseline to showcase the ability of our layered structure. LEX-LSTM is generated by replacing LEX-GAN’s Greplace with a LSTM model. The performance of LEX-LSTM is added in Table 1.
> 3.	The limitation and error cases were briefly discussed in Appendix. We follow the reviewer’s suggestion and add a limitation analysis in Section 5.3. We fixed the reference issue.
>
> We appreciate the important and inspiring comments and suggestions from the reviewer. In addition to the response to the reviews and the modification of the manuscript, we would like to further provide a gentle summary of the contributions of our work:
> 1.	We would like to clarify the difference between explainable rumor detection and explainable fake news detection. Explainable fake news detection has been studied in the literature. However, extension of existing explainable fake news detection works into rumor detection cannot be done because a rumor is defined as an unverified information at the time of posting. Hence, a verified news database cannot be established for explainable rumor detection. To the best of our knowledge, LEX-GAN is the first explainable rumor detection work with demonstrated high accuracy.
> 2.	We would like to provide a quick review of rumor detection and state the strength of our work. Putting aside the explainability, rumor detection has been studied for decades. However, the accuracy of state-of-the-art methods is not promising, which reflects the difficulty of this problem. As we mentioned in the manuscript, to the best of our knowledge, one of the state-of-the-art works GAN-GRU [1] reaches the highest accuracy of 78.1% on a bench-marking rumor dataset PHEME. The average accuracy of other state-of-the-art works on this dataset is around 70%. Our proposed LEX-GAN outperforms all the baselines and achieves 82.4% in terms of accuracy.
> 3.	In addition to rumor detection, we also provide a set of gene mutation detection experiments as an extended application of the proposed LEX-GAN, which showcases the text mining and textual mutation detection power of our framework. We believe our proposed framework could be exploited to make contributions to other domains.
>
> [1] Ma, Jing, Wei Gao, and Kam-Fai Wong. "Detect Rumors on Twitter by Promoting Information Campaigns with Generative Adversarial Learning." The World Wide Web Conference. ACM, 2019.

---

### Official Review · AnonReviewer2 · 2019-10-29
**Official Blind Review #2**

**Rating:** 1

**Review:**

The paper presents a method for detecting rumours in text. Early on in the paper, the authors claim that this method:
1) is more accurate in rumour detection than prior work,
2) can provide explainability, and
3) does not need labelled data because it is trained on synthetic "non-rumour looking rumours".

All three of these statements are problematic.
The experimental evaluation uses a small dataset of rumour classification (about 15000 tweet related to 14 news topics) and an even smaller dataset of gene classification. The rationale is to use the gene classification task as a proxy for rumour detection. This is not valid. The gene classification task does not contribute to the evaluation of the rumour detection method. The rumour classification dataset is relatively small, but even more importantly, the experimental results on that dataset are not thoroughly analysed, for instance through an ablation test.

Explainability is not evaluated experimentally, nor formally proven.

The claim that the method does not need labelled data because it is trained on synthetic "non-rumour looking rumours" is shaky, because 1) one could train the method on labelled data, and 2) it is not clear how "non-rumour looking rumours"  are guaranteed in the synthesis phase (how are they defined? how are they evaluated to be "non-rumour looking rumours"? etc).

Note that there is no definition of what sort of data representation corresponds to a "rumour" in the paper.


**Experience Assessment:**

I have published one or two papers in this area.

**Review Assessment: Checking Correctness Of Derivations And Theory:**

I assessed the sensibility of the derivations and theory.

**Review Assessment: Checking Correctness Of Experiments:**

I assessed the sensibility of the experiments.

**Review Assessment: Thoroughness In Paper Reading:**

I made a quick assessment of this paper.

---

> ### Author Response · Authors · 2019-11-11
> **Response to Review #2 Part I**
>
> We thank the reviewer’s suggestions.
> 1.	The PHEME dataset we used in rumor detection task is a state-of-the-art rumor dataset, unanimously used in a lot of prior works. Here we select some recent publications that used PHEME dataset:
> a.	Han, Sooji, Jie Gao, and Fabio Ciravegna. "Data Augmentation for Rumor Detection Using Context-Sensitive Neural Language Model With Large-Scale Credibility Corpus." (ICLR 2019).
> b.	Zhang, Qiang, et al. "Reply-Aided Detection of Misinformation via Bayesian Deep Learning." The World Wide Web Conference. ACM, 2019.
> c.	Nguyen, Duc Minh, et al. "Fake news detection using deep markov random fields." Proceedings of the 2019 Conference of the North American Chapter of the Association for Computational Linguistics: Human Language Technologies, Volume 1 (Long and Short Papers). 2019.
> d.	Bondielli, Alessandro, and Francesco Marcelloni. "A survey on fake news and rumour detection techniques." Information Sciences 497 (2019): 38-55.
> e.	Conforti, Costanza, Mohammad Taher Pilehvar, and Nigel Collier. "Towards Automatic Fake News Detection: Cross-Level Stance Detection in News Articles." Proceedings of the First Workshop on Fact Extraction and VERification (FEVER). 2018.
> f.	Li, Jing, et al. "A joint model of conversational discourse and latent topics on microblogs." Computational Linguistics 44.4 (2018): 719-754.
> g.	Zubiaga, Arkaitz, et al. "Analysing how people orient to and spread rumours in social media by looking at conversational threads." PloS one 11.3 (2016): e0150989.
> Rumor detection and gene mutation detection are two independent evaluation case studies and are not used as one proxy for the other. While initially we were not aware of a larger gene dataset, we now added a set of experiments with a larger benchmarking gene dataset. We present the results in Section 5.2. We follow the reviewer’s suggestion and added one more set of experiments in rumor detection task that uses leave-one-out rule to test the generalization ability of the model. Our leave-one-out rule works as follows: train the model on some news and test on another news, which essentially provide a realistic testing environment. The experimental results of rumor detection task under leave-one-out rule are shown in Table 1.
> 2.	The explainability of LEX-GAN is evaluated experimentally in both rumor detection and gene mutation detection tasks by reporting Dexplain’s performance. In rumor detection task, Dexplain achieves 80.42% and 81.23% macro-f1 on PHEME’v5 and PHEME’v9, respectively. In table 2, Dexplain’s prediction of suspicious statements in rumors is reported. The explainability of LEX-GAN in rumor detection task is therefore addressed. In gene mutation detection task, Dexplain achieves 89.49% macro-f1 score. In table 4, Dexplain’s prediction of gene mutation is shown and therefore addresses the explainability of LEX-GAN in gene mutation task. Additional results of Dexplain can be found in appendix.
> 3.	We would like to clarify that we didn’t claim LEX-GAN doesn’t need labelled data. We use both real data and generated data to train the model. Rumor is complicated and hard to distinguish since there is no uniform data representation that corresponds to a rumor. LEX-GAN is designed learn the complicated high-dimensional rumor representation. Generated data are used to augment the available dataset and therefore enhance the detection ability of LEX-GAN.

---

> > ### Author Response · Authors · 2019-11-11
> > **Response to Review #2 Part II**
> >
> > We appreciate the important and inspiring comments and suggestions from the reviewer. In addition to the response to the reviews and the modification of the manuscript, we would like to further provide a gentle summary of the contributions of our work:
> > 1.	We would like to clarify the difference between explainable rumor detection and explainable fake news detection. Explainable fake news detection has been studied in the literature. However, extension of existing explainable fake news detection works into rumor detection cannot be done because a rumor is defined as an unverified information at the time of posting. Hence, a verified news database cannot be established for explainable rumor detection. To the best of our knowledge, LEX-GAN is the first explainable rumor detection work with demonstrated high accuracy.
> > 2.	We would like to provide a quick review of rumor detection and state the strength of our work. Putting aside the explainability, rumor detection has been studied for decades. However, the accuracy of state-of-the-art methods is not promising, which reflects the difficulty of this problem. As we mentioned in the manuscript, to the best of our knowledge, one of the state-of-the-art works GAN-GRU [1] reaches the highest accuracy of 78.1% on a bench-marking rumor dataset PHEME. The average accuracy of other state-of-the-art works on this dataset is around 70%. Our proposed LEX-GAN outperforms all the baselines and achieves 82.4% in terms of accuracy.
> > 3.	In addition to rumor detection, we also provide a set of gene mutation detection experiments as an extended application of the proposed LEX-GAN, which showcases the text mining and textual mutation detection power of our framework. We therefore believe our proposed framework could be exploited to make contributions to other domains.
> >
> > [1] Ma, Jing, Wei Gao, and Kam-Fai Wong. "Detect Rumors on Twitter by Promoting Information Campaigns with Generative Adversarial Learning." In The World Wide Web Conference, pp. 3049-3055. ACM, 2019.

---

### Official Review · AnonReviewer4 · 2019-11-01
**Official Blind Review #4**

**Rating:** 8

**Review:**

In this paper, the authors proposed an interesting model to solve rumor detection problem. The LEX-GAN model takes the advantage of GAN in generating high-quality fake examples by substitute few works in original tweets.  It achieved excellent performance on two kinds of dataset.

The term ‘layered’ was a little confusing to me at the very beginning, though it is strengthened in many places around the paper. Maybe the author could use some other word to better summarize the two layers.

Another question is about the extended dataset with generated data, are they generated using the same distribution from G of the final model? What the result would it be if we use real, out-of-domain data?

I would like to see this paper accepted to motivate future works on fake news detection and rumor detection..


**Experience Assessment:**

I have read many papers in this area.

**Review Assessment: Checking Correctness Of Derivations And Theory:**

I assessed the sensibility of the derivations and theory.

**Review Assessment: Checking Correctness Of Experiments:**

I assessed the sensibility of the experiments.

**Review Assessment: Thoroughness In Paper Reading:**

I read the paper at least twice and used my best judgement in assessing the paper.

---

> ### Author Response · Authors · 2019-11-11
> **Response to Review #4**
>
> We thank the reviewer for these important comments.
> As for the term “layered” we used it because in many applications, such as protocols and standards, the term layered is used to imply the breakdown of the steps of the work in multiple steps, called layers. E.g., the TCP/IP layers, etc. We will think and do some more search to see whether a better term could replace “layered”. About the extended dataset, they are generated using the same G in order to ensure the fairness of the comparison between LEX-GAN and all the baselines. We added one more set of experiments that use leave-one-out rule to test the generalization ability of the model. Leave-one-out rule works as follows: train the model on some news and test on another news, which essentially provide a realistic testing environment that contains real, out-of-domain data. The experimental results of rumor detection task under leave-one-out rule are shown in Table 1.
>
> We appreciate the important and inspiring comments and suggestions from the reviewer. In addition to the response to the reviews and the modification of the manuscript, we would like to further provide a gentle summary of the contributions of our work:
> 1.	We would like to clarify the difference between explainable rumor detection and explainable fake news detection. Explainable fake news detection has been studied in the literature. However, extension of existing explainable fake news detection works into rumor detection cannot be done because a rumor is defined as an unverified information at the time of posting. Hence, a verified news database cannot be established for explainable rumor detection. To the best of our knowledge, LEX-GAN is the first explainable rumor detection work with demonstrated high accuracy.
> 2.	We would like to provide a quick review of rumor detection and state the strength of our work. Putting aside the explainability, rumor detection has been studied for decades. However, the accuracy of state-of-the-art methods is not promising, which reflects the difficulty of this problem. As we mentioned in the manuscript, to the best of our knowledge, one of the state-of-the-art works GAN-GRU [1] reaches the highest accuracy of 78.1% on a bench-marking rumor dataset PHEME. The average accuracy of other state-of-the-art works on this dataset is around 70%. Our proposed LEX-GAN outperforms all the baselines and achieves 82.4% in terms of accuracy.
> 3.	In addition to rumor detection, we also provide a set of gene mutation detection experiments as an extended application of the proposed LEX-GAN, which showcases the text mining and textual mutation detection power of our framework. We therefore believe our proposed framework could be exploited to make contributions to other domains.
> [1] Ma, Jing, Wei Gao, and Kam-Fai Wong. "Detect Rumors on Twitter by Promoting Information Campaigns with Generative Adversarial Learning." In The World Wide Web Conference, pp. 3049-3055. ACM, 2019.

---

### Official Review · AnonReviewer3 · 2019-11-04
**Official Blind Review #3**

**Rating:** 1

**Review:**

Three strengths:
1. This paper has been well written and easy to follow. Adequate details have been provided to help easily reproduce the experimental results.
2. The technical part is sound - the authors apply GAN for rumor detection and propose to use model-where and model-replace to extend conventional GAN models.
3. Experiments are conducted on real-world data.

Weaknesses:
1. Contributions of novelty are limited. The idea of using GAN to detect misinformation such as rumors and fake news has been studied in the literature several times, and the proposed method does not differ from them significantly. The problem of explainable rumor and fake news detection has also been well studied. Therefore, this piece of work is more a marginal extension of existing solutions.
2. The technical solution can be very limited. The generator can only manipulate content by replacing something from a true statement. The hidden assumption that misinformation is mostly generated by replacing some word definitely underestimates the complicated nature of fake news/rumor detection problem. If the assumption holds, the rumor detection problem can be easily done by collecting and comparing against true statements.
3. The limited experimental results cannot resolve my concerns. The rumor dataset is very small for a typical deep learning model. I am also curious about how many rumors in the dataset are generated by replacing words.


**Experience Assessment:**

I have published in this field for several years.

**Review Assessment: Checking Correctness Of Derivations And Theory:**

I assessed the sensibility of the derivations and theory.

**Review Assessment: Checking Correctness Of Experiments:**

I carefully checked the experiments.

**Review Assessment: Thoroughness In Paper Reading:**

I read the paper thoroughly.

---

> ### Author Response · Authors · 2019-11-11
> **Response to Review #3 Part I**
>
> We thank the reviewer for all the insightful comments.
> 1.	We would like to clarify the novel contributions of our work:
> a.	We are not aware of any research on explainable rumor detection. We agree with the reviewer that explainable misinformation and fake news detection has been studied in the literature. However, extension of existing explainable fake news detection works into rumor detection cannot be done because a rumor is defined as an unverified information at the time of posting. Hence, a verified news database cannot be established for explainable rumor detection. To the best of our knowledge, LEX-GAN is the first explainable rumor detection work with demonstrated high accuracy.
> b.	We further clarify the distinction between explainable fake news work and rumor detection. In explainable fake news or misinformation detection, a larger verified background knowledge set and training dataset are needed to provide explainability. For example, in work [1], a verified news set needs to be collected to provide explainability. In work [2-3], explainability relies on the comments of the fake news and/or metainformation of users. Without these additional information and verified veracity of sentences, the explainable fake news detection problem becomes a rumor detection problem. However, these explainable fake new detection methods cannot be deployed directly because the shortage of these additional data. LEX-GAN, on the contrary, proposes a solution to explainable rumor detection without requesting additional background information and can easily be extended to explainable fake news detection, if these additional data are available.
> c.	Putting aside the explainability, rumor detection has been studied for decades. However, the accuracy of state-of-the-art methods is not promising, which reflects the difficulty of this problem. As we mentioned in the manuscript, to the best of our knowledge, one of the state-of-the-art works GAN-GRU [4] reaches the highest accuracy of 78.1% on a bench-marking rumor dataset PHEME. The average accuracy of other state-of-the-art works on this dataset is around 70%. Our proposed LEX-GAN outperforms all these prior approaches on state-of-the-art benchmarks and achieves 82.4% in terms of accuracy.
> d.	In addition to rumor detection, we also provide a set of gene mutation detection experiments as an extended application of the proposed LEX-GAN, which showcases the text mining and textual mutation detection power of our framework. We believe our proposed framework could be exploited and make contributions to other domains.
> 2.	Content manipulation is a way of augmenting the available data, hence improves the detection ability of LEX-GAN. We did not assume that rumor is mostly generated by replacing some words in a sentence. We acknowledge the complicated nature of the rumor and LEX-GAN captures it by training on both original complicated rumors and generated rumors. Here we provide two examples to demonstrate the rumor detection power of LEX-GAN compared to baselines.
> a.	A rumor example that are correctly detected by LEX-GAN but incorrectly detected by other baselines.
> e.g. “who's your pick for worst contribution to sydneysiege mamamia uber or the daily tele”
> LEX-GAN predicted suspicious words (in parenthesis):
> “(who’s) your pick for worst (contribution) to sydneysiege (mamamia uber) or the daily tele”
> LEX-GAN score: 0.1579
> Baseline CNN score: 0.9802
> Baseline LSTM score: 0.9863
> Baseline VAE-CNN score: 0.4917
> Baseline VAE-LSTM score: 0.5138
> Score 0 and 1 represent the model predicts the sentence as rumor and non-rumor, respectively.
> b.	A non-rumor example that are correctly detected by LEX-GAN but incorrectly detected by other baselines, as a rumor, i.e., with low scores.
> e.g. “glad to hear the sydneysiege is over but saddened that it even happened to begin with my heart goes out to all those affected 💜”
> LEX-GAN score: 0.8558
> Baseline CNN score: 0.0029
> Baseline LSTM score: 0.1316
> Baseline VAE-CNN score: 0.6150
> Baseline VAE-LSTM score: 0.4768
> As we can see in example a, LEX-GAN provides a very low score for a rumor, while other baselines all generated relatively high scores, and even detect it as non-rumor. This is a very difficult example since from the sentence itself, we as human rumor detection agents even cannot pick the suspicious parts confidently. However, LEX-GAN gives a reasonable prediction and shows that it has the ability to understand and analyze complicated rumors. In example b, a non-rumor sentence gains a high score from LEX-GAN, but several relatively low scores from the baselines. This example again confirms that our proposed LEX-GAN indeed captures the complicated nature of rumors and non-rumors.

---

> > ### Author Response · Authors · 2019-11-11
> > **Response to Review #3 Part II**
> >
> > 3.	The dataset we used in the evaluation of LEX-GAN against prior work on rumor detection task is a state-of-the-art rumor dataset. In addition to the real rumors in the datasets, we generated in total around 8000 rumors. Here we select some very recent publications that used PHEME dataset:
> > a.	Han, Sooji, Jie Gao, and Fabio Ciravegna. "Data Augmentation for Rumor Detection Using Context-Sensitive Neural Language Model With Large-Scale Credibility Corpus." (ICLR 2019).
> > b.	Zhang, Qiang, et al. "Reply-Aided Detection of Misinformation via Bayesian Deep Learning." The World Wide Web Conference. ACM, 2019.
> > c.	Nguyen, Duc Minh, et al. "Fake news detection using deep markov random fields." Proceedings of the 2019 Conference of the North American Chapter of the Association for Computational Linguistics: Human Language Technologies, Volume 1 (Long and Short Papers). 2019.
> > d.	Bondielli, Alessandro, and Francesco Marcelloni. "A survey on fake news and rumour detection techniques." Information Sciences 497 (2019): 38-55.
> > e.	Conforti, Costanza, Mohammad Taher Pilehvar, and Nigel Collier. "Towards Automatic Fake News Detection: Cross-Level Stance Detection in News Articles." Proceedings of the First Workshop on Fact Extraction and VERification (FEVER). 2018.
> > f.	Li, Jing, et al. "A joint model of conversational discourse and latent topics on microblogs." Computational Linguistics 44.4 (2018): 719-754.
> > g.	Zubiaga, Arkaitz, et al. "Analysing how people orient to and spread rumours in social media by looking at conversational threads." PloS one 11.3 (2016): e0150989.
> >
> >
> > [1] Yang, Fan, Shiva K. Pentyala, Sina Mohseni, Mengnan Du, Hao Yuan, Rhema Linder, Eric D. Ragan, Shuiwang Ji, and Xia Ben Hu. "XFake: Explainable Fake News Detector with Visualizations." In The World Wide Web Conference, pp. 3600-3604. ACM, 2019.
> > [2] Cui, Limeng, Kai Shu, Suhang Wang, Dongwon Lee, and Huan Liu. "dEFEND: A System for Explainable Fake News Detection." In Proceedings of the 28th ACM International Conference on Information and Knowledge Management, pp. 2961-2964. ACM, 2019.
> > [3] Ruchansky, Natali, Sungyong Seo, and Yan Liu. "Csi: A hybrid deep model for fake news detection." In Proceedings of the 2017 ACM on Conference on Information and Knowledge Management, pp. 797-806. ACM, 2017.
> > [4] Ma, Jing, Wei Gao, and Kam-Fai Wong. "Detect Rumors on Twitter by Promoting Information Campaigns with Generative Adversarial Learning." In The World Wide Web Conference, pp. 3049-3055. ACM, 2019.

---

> ### Comment · AnonReviewer3 · 2019-11-13
> **Response to the authors' comments**
>
> Thanks for the detailed response and I appreciate the efforts of improving the manuscript. However, there's a clear consensus among most reviewers that the novelty is limited, and it is unclear how this method significantly differentiates from existing work in the fake-news and misinformation detection domain - I understand there is a difference such as the word manipulation, but it seems to be incremental rather than originally novel.
>
> The paper has been really well written and easy to follow. I would suggest the authors investigate more about the behaviors of rumor authors and make it convincing in a quantitative way that detecting word manipulation is key in this area.

---

> > ### Author Response · Authors · 2019-11-15
> > **Response to reviewer3's comments**
> >
> > We appreciate the reviewer's effort and time and giving us the opportunity to clarify the novelty of our manuscript. We would like to clarify that the novelty of our manuscript is not about a mathematical formula breakthrough, but rather about a framework we designed for extracting information from text without consulting a background information dataset and metainformation of the text. We agree that the components we used in this framework, such as LSTM, GRU, CNN, are not new, but the layered GAN architecture we proposed for rumor detection is novel and high-performance. The major contribution of this work is the delicate way we employ to design an architecture out of these components and the design of this high-performance explainable rumor detector.
> >
> > We are aware of only one GAN-based rumor detection work [1]. This work uses GAN to convert rumor to non-rumor, and non-rumor to rumor based on patterned words. For example, a rumor contains some obvious words like “fake news”, “is this true?”, "not believe”, “not sure”, etc. To the best of our knowledge, this work achieves the highest performance, 78.1% accuracy, on PHEME dataset, which is way higher than state-of-the-art works. However, this work doesn’t provide explainability, and couldn’t be extended to provide reasonable explainability. For example, if the patterned words don’t appear in the text, then this model is not capable to provide reasonable explainations since it doesn’t deeply capture the semantic meaning of the text. This is the fundamental reason why our LEX-GAN outperforms this model. LEX-GAN is designed for providing explainality and detecting real-world rumors. Our major goal is to differentiate real-world rumors and non-rumors, but not to detect word manipulation in synthetic data. We train LEX-GAN by GAN techniques to let it gain the ability of understanding the text and extracting meaningful information from the text for rumor detection. Word manipulation is a way of augmenting data. It enables LEX-GAN to extract suspicious parts in the sentences and better recognize the rumor/non-rumor patterns.
> >
> > Without explainability, rumor detection methods and fake news detection methods are quite similar and can be extended to perform similar tasks.
> >
> > With explainability, however, the methods for tackling these two are quite different. As stated in our previous reply, the major difference is the requirement of verified news database and metainformation of the users and posts.
> >
> > Explainability is provided by the discriminator Dexplain through the adversarial training process. Different than ordinary adversarial training in GAN, we use manipulated text instead of text generated from scratch to enhance the ability of discriminators. The generators manipulate the text by taking not only Dexplian’s feedback, but also Dclassify’s feedback into consideration. This approach provides the high accuracy of LEX-GAN. Intelligent word manipulation generates an augmented dataset which not only helps the discriminators to extract meaningful rumor/non-rumor patterns, but also exposes the discriminators in an environment full of statements with unseen patterns. This procedure strengthens the rumor classification of LEX-GAN and results in on average 26.85% macro-f1 outperformance under PEHME.
> >
> > [1] Ma, Jing, Wei Gao, and Kam-Fai Wong. "Detect Rumors on Twitter by Promoting Information Campaigns with Generative Adversarial Learning." In The World Wide Web Conference, pp. 3049-3055. ACM, 2019.

---

### Decision · Program_Chairs · 2019-12-19

**Decision:**

Reject

**Comment:**

The paper is well-written and presents an extensive set of experiments. The architecture is a simple yet interesting attempt at learning explainable rumour detection models. Some reviewers worry about the novelty of the approach, and whether the explainability of the model is in fact properly evaluated. The authors responded to the reviews and provided detailed feedback. A major limitation of this work is that explanations are at the level of input words. This is common in interpretability (LIME, etc), but it is not clear that explanations/interpretations are best provided at this level and not, say, at the level of training instances or at a more abstract level. It is also not clear that this approach would scale to languages that are morphologically rich and/or harder to segment into words. Since modern approaches to this problem would likely include pretrained language models, it is an interesting problem to make such architectures interpretable.